# Development of an Orodispersible Film Containing Stabilized Influenza Vaccine

**DOI:** 10.3390/pharmaceutics12030245

**Published:** 2020-03-08

**Authors:** Yu Tian, Yoshita C. Bhide, Herman J. Woerdenbag, Anke L. W. Huckriede, Henderik W. Frijlink, Wouter L. J. Hinrichs, J. Carolina Visser

**Affiliations:** 1Department of Pharmaceutical Technology and Biopharmacy, University of Groningen, Antonius Deusinglaan 1, 9713 AV Groningen, The Netherlands; y.tian@rug.nl (Y.T.); y.c.bhide@umcg.nl (Y.C.B.); h.j.woerdenbag@rug.nl (H.J.W.); h.w.frijlink@rug.nl (H.W.F.); j.c.visser@rug.nl (J.C.V.); 2Department of Medical Microbiology and Infection Prevention, University Medical Center Groningen, University of Groningen, Antonius Deusinglaan 1, 9713 AV Groningen, The Netherlands; a.l.w.huckriede@umcg.nl

**Keywords:** whole inactivated influenza virus vaccine, orodispersible films, stabilization, trehalose, pullulan, β-galactosidase, hemagglutination

## Abstract

Most influenza vaccines are administered via injection, which is considered as user-unfriendly. Vaccination via oral cavity using an orodispersible film (ODF) might be a promising alternative. To maintain the antigenicity of the vaccine during preparation and subsequent storage of these ODFs, sugars such as trehalose and pullulan can be employed as stabilizing excipients for the antigens. In this study, first, β-galactosidase was used as a model antigen. Solutions containing β-galactosidase and sugar (trehalose or trehalose/pullulan blends) were pipetted onto plain ODFs and then either air- or vacuum-dried. Subsequently, sugar ratios yielding the highest β-galactosidase stability were used to prepare ODFs containing H5N1 whole inactivated influenza virus vaccine (WIV). The stability of the H5N1 hemagglutinin was assessed by measuring its hemagglutination activity. Overall, various compositions of trehalose and pullulan successfully stabilized β-galactosidase and WIV in ODFs. WIV incorporated in ODFs showed excellent stability even at challenging storage conditions (60 °C/0% relative humidity or 30 °C/56% relative humidity) for 4 weeks. Except for sugars, the polymeric component of ODFs, i.e., hypromellose, possibly improved stability of WIV as well. In conclusion, ODFs may be suitable for delivering of WIV to the oral cavity and can possibly serve as an alternative for injections.

## 1. Introduction

Influenza is one of the most serious infectious diseases responsible for high morbidity and mortality [1]. Vaccination is an efficient method to control yearly influenza epidemics and may help to combat occasional pandemics [2]. Except for Flumist^®^, a live attenuated influenza vaccine that is administered intranasally, other marketed influenza vaccines are administered via the parenteral route. The disadvantage of injections, however, is that they need to be administered by trained health care personnel. In addition, needle stick injuries may occur by which infectious diseases can be transmitted. Finally, compliance is jeopardized for individuals with needle phobia.

Vaccination by the buccal or sublingual route is non-invasive, user-friendly and safe, and may be an attractive alternative delivery route. Buccal or sublingual vaccination is easy for self-administration [3,4,5,6]. Moreover, buccal or sublingual administration of influenza vaccine has been shown to induce mucosal immune responses in lungs, which plays an important role in neutralizing influenza virus directly at its portal of entry [7,8]. Antigen presenting cells (APCs), e.g., dendritic cells and Langerhans cells, are abundantly present in the mucosa of the oral cavity. These APCs as well as the mucosa-associated lymphoid tissue (MALT) in the oral cavity, e.g., Waldeyer’s ring and palatine tonsils [9], enhance the capture and uptake of the antigen. Upon antigen uptake, APCs migrate to draining lymph nodes in buccal or sublingual area where they present the processed antigen to CD4 and CD8 T cells. Several studies have shown that immunization by delivery of antigens to the oral cavity is feasible as described in the reviews of Kweon et al. [6] and Kraan et al. [10]. Oberoi et al. [11] administered split influenza vaccine adjuvanted with CRX-601 in liposomes via the sublingual route, which elicits strong mucosal immune responses, while administration by intramuscular route did not produce any mucosal immune responses. Murugappan et al. [12] investigated a prime/boost vaccination regimen using whole inactivated influenza virus vaccine (WIV), i.e., immunization by sublingual priming, which showed higher mucosal immune responses in nose and lungs than immunization with intramuscular priming and boosting.

Orodispersible films (ODFs) would be ideal dosage forms to achieve buccal or sublingual administration. The aim of the present study was to develop an ODF for delivery of an influenza vaccine to the oral cavity. ODFs are generally prepared by solvent casting techniques. Usually, a solution containing film forming polymers, other excipients (e.g., plasticizer) and the active pharmaceutical ingredient (API), are cast on a suitable substrate and then dried. However, such production method would imply quite some waste of vaccine as the edges of the films left after cutting films into the desired shape are usually discarded. To overcome this disadvantage, we used a plain ODF, developed by us in a previous study [13], onto which the vaccine solution containing stabilizers was pipetted followed by air- or vacuum-drying. This procedure enables precise and low volume antigen deposition. As antigen, we selected WIV because it contains all the structural viral proteins and functions as a promising basis for the influenza vaccines. WIV has stronger immunogenic properties than split virus or subunit vaccine, as it retains the single stranded viral genomic RNA which can be recognized by Toll-like receptors and acts as an intrinsic adjuvant [14,15].

Like other biopharmaceuticals, hemagglutinin (HA), the major antigenic component of WIV, is prone to degradation when stresses occur during processing (e.g., shear and dehydration stresses) and storage. It is well known that biopharmaceuticals can be stabilized during drying and subsequent storage by incorporating them in a sugar matrix in the glassy state [16,17,18]. Indeed, in previous studies we have shown that sugars can be used to stabilize WIV during drying and subsequent storage [12,19]. Furthermore, we have shown that a combination of trehalose and pullulan can be used to stabilize proteins in ODFs [19]. It was hypothesized that the disaccharide trehalose, having a low molecular weight, can provide a compact molecular coating around protein. However, trehalose has a relatively low glass transition temperature, which might lead to crystallization especially when exposed to high relative humidities. On the other hand, the polysaccharide pullulan has a very high glass transition temperature, hence, blends of trehalose and pullulan seem to possess preferred physico-chemical properties to enhance the stability of proteins [17]. Therefore, we evaluated whether trehalose only, blends of trehalose and pullulan or both can be used to stabilize WIV during incorporation in ODFs and subsequent storage.

Due to limited availability of WIV, we first used β-galactosidase as a model antigen to select the best formulations [20]. The enzymatic activity of β-galactosidase after incorporation in ODFs and subsequent storage was investigated. In addition, the uniformity of weight and thickness, disintegration time and mechanical properties of the ODFs were evaluated. The best formulations were selected to incorporate WIV in the ODFs. The stability of WIV was determined after processing and after storage using the hemagglutination capacity of WIV as a read out for vaccine stability.

## 2. Materials and Methods

### 2.1. Materials

Pullulan (average molecular weight 200–300 kDa) and trehalose were kind gifts from Hayashibara (Okayama, Japan). β-galactosidase (molecular weight 540 kDa) was obtained from Sorachim (Lausanne, Switzerland). Bovine serum albumin (BSA) was obtained from Sigma-Aldrich (St. Louis, MO, USA). Hypromellose (HPMC, Methocel E3 premium LV) was kindly received from Colorcon (Kent, UK). Glycerol 85%, carbomer 974P, disodium ethylenediaminetetraacetate (disodium edetate) and trometamol were obtained from Fagron (Capelle aan den IJssel, The Netherlands). All other excipients and chemicals were of analytical grade.

### 2.2. Virus Preparation

NIBRG-23, a reassortant of A/turkey/Turkey/1/2005 (H5N1) and A/PR/8/34 (H1N1) was obtained from the National Institute of Biological Standards and Controls, Potters Bar, United Kingdom. The virus was produced in the allantoic cavity of 11-day-old embryonated hens’ eggs. The virus was purified and inactivated as described by Tomar et al. to obtain WIV [21,22]. WIV was derived from the same strain as previously published [21,22]. The size of WIV (around 185 nm) together with other physico-chemical properties of WIV were described in these papers.

### 2.3. Preparation of the Casting Solution and ODFs

The casting solution as developed by Visser et al. was used to prepare plain ODFs [23]. HPMC, carbomer 974P, disodium edetate, trometamol, and glycerol 85% were added to water under constant stirring at 1000 rpm. After complete dissolution, the entrapped air bubbles were removed from the solution by stirring at low speed (100 rpm) for an additional 48 h. The solution was cast onto a release liner (Primeliner 410/36, Loparex, Apeldoorn, The Netherlands) at the speed of 10 mm/s by using a coatmaster (Erichsen, Hemer, Germany) and with a casting height of 1000 μm. The ODFs were dried on the plate of the coatmaster at 30 °C for 10 min before pipetting the sugar solutions containing β-galactosidase or WIV onto them.

### 2.4. Preparation of Sugar Solutions with Antigen and Antigen Incorporated ODFs

Various solutions with either β-galactosidase or WIV incorporated were prepared in 2 mM HEPES buffer pH 7.4 as listed in Table 1 and Table 2, respectively. In some formulations, BSA was incorporated to reduce potential absorption of antigen onto the wall of the glass container (beaker) during preparation or onto the pipetting tips. Methylene blue was used to make the antigen dots visible.

Antigen solutions were pipetted onto the ODFs by using a grid pattern at a volume of 6 μL. The ODFs were air-dried for 4 h at 30 °C (50–60% relative humidity) on the plate of the coatmaster or vacuum-dried for 24 h (using a Christ Epsilon 2–4 lyophilizer (Salm & Kipp, Breukelen, The Netherlands). The shelf temperature was maintained at 0 °C for 10 min at 0.05 mbar after which the chamber pressure was reduced to 0.03 mbar for 1 h. Subsequently, shelf temperature was increased at 0.05 °C/min to 20 °C while keeping the chamber pressure at 0.03 mbar [24].

The ODFs with β-galactosidase incorporated were punched in squares of 1.8 × 1.8 cm using an Artemio perforator (Artemio, Wavre, Belgium). Each of these ODFs had six β-galactosidase dots and the ODFs were sealed in a plastic bag and stored at −20 °C until characterization. Plain ODFs were compared with β-galactosidase containing ODFs for uniformity of weight and thickness, disintegration, and mechanical properties. ODFs containing two dots of β-galactosidase were put into Eppendorf tubes and stored at −20 °C and used for β-galactosidase enzymatic activity assay within 24 h after preparation. We assumed that β-galactosidase did not lose any activity during 24 h of storage at −20 °C.

ODFs with one dot of WIV containing 5 μg of protein were cut and also put into Eppendorf tubes. To investigate the effects of the excipients of the ODFs on the stability of WIV, WIV solutions were also dried as follows. WIV dots were prepared by pipetting 6 μL of WIV solution directly onto release liner, thus without ODFs, followed by air- or vacuum-drying as described above. The release liner was cut into pieces containing one WIV dot, which were put in Eppendorf tubes. Eppendorf tubes were stored at −20 °C and analyzed within 24 h after preparation by the hemagglutination assay as described below. We assumed that WIV did not lose any activity during 24 h of storage at −20 °C.

### 2.5. Uniformity of Mass and Thickness

Six randomly chosen β-galactosidase containing ODFs of each formulation as well as plain ODFs were weighed individually on an analytical balance. The average weight and weight variation were calculated.

The thickness of ODFs was measured with a micro screw meter (Mitutoyo, Neuss, Germany) at five different points of the ODFs: in the corners and in the middle. The thickness of each film was considered as the average of five points.

### 2.6. Disintegration Time

The disintegration time of ODFs with β-galactosidase incorporated and plain ODFs was determined with an adapted slide frame method as previously published [25]. The ODFs (*n* = 6) were clamped in an arm, which moved up and down at a frequency of 30 ± 1 cycles per min, over a distance of 55 ± 2 mm in a water bath at 37 °C ± 1 °C. The time at which ODFs were completely dissolved was recorded as the disintegration time. The endpoint was judged by visual inspection.

### 2.7. Mechanical Properties

The mechanical properties of ODFs with β-galactosidase incorporated and plain ODFs were analyzed using an Instron series 5500 load frames with a load cell of 100 N (Instron, Norwood, USA) [13,18,25]. ODF were cut into a bone shape according to ISO-527 standard (plastics-determination of tensile properties) (NEN-EN-ISO, 2012). The ODFs (*n* = 8) were fixed between two clamps positioned at a distance of 4 mm. Subsequently, the clamps were moved away from each other with a cross-head speed of 50 mm/min until tearing or breakage of the ODFs. Tensile strength (N/mm^2^) and elongation at break (%) were recorded and automatically calculated by using Instron Merlin (series IX).

### 2.8. Process and Storage Stability Testing

The process and storage stability of β-galactosidase or WIV incorporated in ODFs was investigated. The storage stability was tested under closed vial conditions (0% relative humidity (RH)) at 4 °C, 30 °C and 60 °C, and under open vial conditions at 30 °C and 56% RH, generated by using a saturated sodium bromide solution [17,26]. The enzymatic activity of β-galactosidase and hemagglutination capacity of WIV were assessed immediately after the drying process (process stability) and after 1, 2 and 4 weeks of storage (storage stability). To investigate the storage stability of WIV without ODFs, the hemagglutination capacity of WIV dots was evaluated after 1, 2, 4 and 7 days of storage.

### 2.9. β-Galactosidase Enzymatic Activity Assay

The β-galactosidase activity was determined using a kinetic enzymatic assay, based on the conversion of a colorless substrate, o-nitrophenylgalactoside, into the yellow colored product, o-nitrophenol, by β-galactosidase [17,18]. Firstly, ODFs containing two dots of β-galactosidase were cut into small pieces and dissolved in 1.0 mL of 0.1 M phosphate buffer (pH 7.3), after which samples were diluted fivefold with enzyme diluent solution containing 0.1% BSA and 1 mM MgCl_2_ in 50 mM phosphate buffer. Subsequently, 20 μL samples (*n* = 3) were pipetted into each well of a 96-well microplate (Greiner Bio-One, F shape), followed by 200 μL of 1.4 mM MgCl_2_ in 0.1 M phosphate buffer. The plate was incubated at 37 °C for 10 min. Then, 20 μL of 50 mM substrate o-nitrophenyl-galactoside was added. The absorption was measured at 405 nm for 15 min at 37 °C with 30 s intervals (Synergy HT Microplate Reader, BioTek Instruments, Winooski, VT). The β-galactosidase activity was calculated from the slope of this conversion. All measurements were performed in triplicate.

### 2.10. Hemagglutination Assay

The activity of WIV in ODFs or in dots was assessed by the hemagglutination assay performed as previously described [22,27]. HA glycoprotein at the surface of WIV is able to bind to red blood cells (RBCs) and cause RBCs agglutination. In the absence of virus particles, RBCs precipitate to the bottom of the well by gravity, showing a red-colored dot in the well. In the presence of virus, RBCs clump together and no red dot is formed.

ODFs or WIV dots were dissolved in 0.1 M phosphate buffer (pH 7.3) to obtain a WIV concentration of 0.05 μg/μL. Subsequently, 100 μL of this solution was pipetted into 96-well V bottom plates and two-fold serially diluted. Thereafter, 50 μL of 1.5% guinea pig RBCs were added to each well and hemagglutination titers were read through visual inspection after 2 h incubation at room temperature. Hemagglutination titers were expressed as the log_2_ of the highest dilution where RBCs agglutination occurred. All measurements were performed in triplicate.

### 2.11. Statistical Analysis

The results were statistically analyzed using one-way analysis of variance (ANOVA). A *p*-value < 0.05 was considered as significantly different. *p*-value less than 0.05, 0.01, 0.001, and 0.0001 are denoted by *, **, *** and ****, respectively. The graphs and curve fittings were performed using GraphPad Prism version 6.0 (GraphPad Prism Software, Inc., La Jolla, CA, USA).

## 3. Results and Discussion

β-Galactosidase was selected as a model antigen because it is readily available and relatively unstable [20]. The remaining enzymatic activity of β-galactosidase after incorporation in the ODFs and after subsequent storage was measured. In addition, uniformity of mass and thickness, disintegration and mechanical properties of the ODFs with β-galactosidase incorporated were determined. Subsequently, formulations with the best stability were applied for the incorporation of WIV into ODFs. The stability of WIV incorporated in ODFs and the influence of excipients of ODFs on WIV activity were investigated as well.

### 3.1. ODFs with β-Galactosidase Incorporated

#### 3.1.1. Characterizations of ODFs


(a)Uniformity of Mass and Thickness and DisintegrationAs shown in Table 3 and Table 4, incorporation of β-galactosidase together with sugars in ODFs showed acceptable uniformity of mass and thickness with low standard deviations. All ODFs disintegrated within 30 s, which is recommended by U.S. Food and Drug Administration (FDA) for orally disintegrating tablets [28]. Furthermore, vacuum-dried ODFs showed slightly shorter (but not significantly different) disintegration time than air-dried ODFs.(b)Mechanical PropertiesAs can be seen in Table 3 and Table 4, by the incorporation of sugars, the ODFs had slightly lower tensile strength and elongation at break than plain ODFs (*p* < 0.0001), which means they became more brittle and thus less flexible. Furthermore, ODFs with the highest trehalose concentration (Trehalose (1.0) and BSA-Trehalose (1.0)) had the lowest tensile strength and elongation at break in both air- and vacuum-dried ODFs, which means they were the most brittle and fragile. Additionally, in a previous study, we found that incorporation of increasing amounts of trehalose in ODFs resulted in increasing deterioration of the mechanical properties [18]. This phenomenon can be explained by the low molecular weight of trehalose, which makes it a poor film former. Vacuum-dried ODFs showed a slightly lower (but not significantly different) tensile strength and elongation at break than air-dried ODFs.


Breaks sometimes occurred during handling or during the drying process for ODFs with highest trehalose concentration.

#### 3.1.2. Enzymatic Activity of β-Galactosidase

Due to its ready availability and the straightforward quantitation by determination of its enzymatic activity, β-galactosidase has been widely used as model protein for stabilization by sugars [16,17]. β-Galactosidase formulated with trehalose by freeze-drying was found to be the most stable formulation when compared with other sugars (i.e., dextran 70 kDa, dextran 6 kDa and inulin) [16]. In a study by Lipiäinen et al. [29], melibiose or trehalose was used as stabilizing excipient for β-galactosidase by spray drying, and its activity remained for 30 days at 40 °C.

Immediately after drying, β-galactosidase activity of ODFs formulated with sugars was significantly higher than without sugar, independent of the drying method (Figure 1). The different sugar containing formulations had slight differences in process stability of β-galactosidase. Trehalose (0.5) showed the lowest process stability, 62% for air-drying ODFs and 65% for vacuum-drying ODFs. BSA-Trehalose (1.0) and Trehalose (1.0) exhibited the highest process stability, ending up at around 85% for both air- and vacuum-dried ODFs. Furthermore, the use of BSA in β-galactosidase incorporated sugar solutions seemed to have slightly positive effect on the process stability of β-galactosidase, however, as mentioned, the differences were minor.

Storage at 4 °C/0% RH for 4 weeks had no significant effect on β-galactosidase activity for both air- and vacuum-dried ODFs when sugars were incorporated in the formulation (Figure 1A,B). Although incorporation of β-galactosidase in ODFs without sugar did not result in a decline of enzymatic activity during the first 2 weeks of storage, activity decreased to 20% upon an additional 2 weeks of storage. Therefore, it can be concluded that when properly formulated, storage at 4 °C/0% RH seems a suitable storage condition for ODFs containing β-galactosidase.

During storage at 4 °C/0% RH for 4 weeks, no distinction between the stabilizing capacities of the different sugar containing formulations could be made, because the ODFs were exposed to more challenging storage conditions, i.e., 30 °C/0% RH. During 4 weeks’ storage at this condition, β-galactosidase incorporated in ODFs with both pullulan and trehalose with and without BSA in air-dried ODFs had better storage stability than other formulations, showing a remaining enzymatic activity of 40–50% (Figure 1C,D). For example, air-dried ODFs with the formulations Trehalose (0.5), Trehalose (0.75), Trehalose (1.0) and BSA-Trehalose (1.0) exhibited remaining β-galactosidase activities below 20%, while air-dried ODFs with β-galactosidase incorporated without sugar almost completely lost its activity. Compared to air-dried ODFs, β-galactosidase incorporated in vacuum-dried ODFs generally showed a better storage stability, which might be explained by the fact that the vacuum-drying process (24 h) was longer than the air-drying process (4 h). During the vacuum-drying, the mobility of sugar reduced in a slower rate, which probably contributes to the better encapsulation of β-galactosidase by sugar. After 4 weeks’ storage at 30 °C/0% RH, the remaining β-galactosidase activity was between 20–60% when sugars were present. The two formulations with BSA incorporated in vacuum-dried ODFs showed good storage stability as well, with a remaining β-galactosidase activity of 50–60% after 4 weeks. β-Galactosidase incorporated in vacuum-dried ODFs without sugar showed the lowest storage stability (12%).

### 3.2. ODFs with WIV Incorporated

Based on the results obtained with β-galactosidase, the formulations BSA-Trehalose (0.4) pullulan (0.1), Trehalose (0.4) pullulan (0.1) and BSA-Trehalose (1.0) were applied for incorporation of WIV into ODFs. A WIV solution without sugar was also pipetted onto ODFs as a negative control. WIV dots were prepared to investigate the influence of excipients of ODFs on the stability of WIV.

#### 3.2.1. Biological Activity of WIV Incorporated into ODFs

The biological activity of HA of WIV was determined by the hemagglutination assay. In preliminary studies, we found that WIV incorporated in ODFs together with sugars was stable with a hemagglutination titer which remained constant at around 5 log_2_ during 8 weeks’ storage at 4 °C/0% RH and even at 30 °C/0% RH (Appendix A). Therefore, it was decided to expose WIV containing ODFs to more challenging storage conditions, i.e., 60 °C/0%RH and 30 °C/56% RH.

As shown in Figure 2, immediately after preparation, the hemagglutination titer was significantly higher when WIV was formulated with sugars than without, for both air- and vacuum-dried ODFs (*p* < 0.05). Upon storage for 4 weeks at 60 °C/0% RH, hemagglutination titers were reduced in all samples, while no differences were found between titers upon air- or vacuum-drying. The hemagglutination titers of WIV formulated without sugar decreased from around 4.5 log_2_ immediately after preparation to around 3.5 log_2_ after 4 weeks of storage. No significant difference in storage stability at 60 °C/0% RH was found between the different formulations containing sugars; i.e., they all resulted in reduction of the hemagglutination titer from around 6 log_2_ immediately after preparation of the ODFs to around 5 log_2_ after 4 weeks. When stored under high moisture conditions, i.e., 30 °C/56% RH, the stability of WIV showed a similar trend as for 60 °C/0% RH. WIV formulated without sugar only showed a slight decrease of hemagglutination titer during storage: i.e., from around 4.5 log_2_ immediately after preparation to around 4 log_2_ after 4 weeks. All formulations with sugars showed similar stabilities upon storage. The addition of pullulan did not significantly improve WIV stability since WIV formulated with only trehalose or with blends of trehalose and pullulan did not show a significant difference in its hemagglutination titer. In other words, WIV can be stabilized either by only trehalose or by a blend of pullulan and trehalose in ODFs, and remains quite stable for at least 4 weeks at both 60 °C/0% RH and 30 °C/56% RH.

Besides, the differences of the stability of WIV among various sugar containing formulations were not as pronounced as for β-galactosidase incorporated in ODFs. Furthermore, WIV incorporated in ODFs appeared to be much more stable than β-galactosidase incorporated in ODFs. Apparently, WIV has a higher intrinsic stability. A high stability of WIV incorporated in sugar glasses was also shown by Geeraedts et al. [19] who found a reduction of the hemagglutination titer of WIV derived from H5N1 influenza virus (NIBRG-14) freeze-dried in the presence of trehalose or inulin from 11 log_2_ to 8 log_2_ during 3 months of storage at 40 °C. Murugappan et al. [30] found that the hemagglutination titer of WIV derived from A/Hir/H3N2 influenza virus spray freeze-dried in the presence of inulin, dextran or dextran/trehalose mixture remained constant at around 10 log_2_ and did not change during 3 months storage at 40 °C.

#### 3.2.2. Biological Activity of WIV without ODFs

In the previous part of this study, WIV was incorporated in ODFs by pipetting the sugar solution with antigen onto plain ODFs followed by air- or vacuum-drying. As the plain ODFs were composed of readily water-soluble components (i.e., predominately HPMC), the pipetted solution will locally and partially dissolve these. As a consequence, after drying the antigen will not only be encapsulated by the sugars when present in the pipetting solution, but also by ODF components (HPMC), which may affect stability [31,32]. In order to investigate the stabilizing or destabilizing effects of ODF components, WIV dots that were prepared by pipetting sugar/WIV solution directly onto the release liner were analyzed. The samples were stored at 60 °C/0% RH or 30 °C/56% RH and after different time intervals, the hemagglutination titers of WIV were determined.

As shown in Figure 3, the process stability of WIV dots (around 3 log_2_) was significantly lower than that of WIV incorporated in ODFs. Upon storage, WIV without ODFs were more sensitive to temperature than to humidity. Both air- and vacuum-dried WIV dots without sugar fully lost their activity after 2 days of storage at 60 °C/0% RH. In contrast, air-dried WIV dots with sugar had a remaining hemagglutination titer of around 1 log_2_ after 7 days of storage at 60 °C/0% RH. Vacuum- dried WIV dots showed similar stability as air-dried WIV samples upon storage. At 30 °C/56% RH storage condition, the storage stability of WIV dots was slightly higher than that at 60 °C/0% RH. WIV dots formulated without sugar had lowest hemagglutination titer as well, ending up at 0.5 log_2_ after 7 days of storage. The storage stability of WIV dots formulated with sugars decreased gradually and reached around 2 log_2_ after 7 days.

Compared with WIV incorporated in ODFs, WIV dots had significantly lower hemagglutination titers. The dramatic loss of WIV activity indicates the significant protective ability of ODFs on WIV. The major component of ODFs’ HPMC probably contributes to improve stability of WIV. Furthermore, the slight dissolution of ODF on the spot of pipetting may cause the encapsulation of WIV, which encloses it not only by sugar but also by the components of ODFs, as a protection layer at the surface of WIV against higher temperatures and moisture. Consequently, not only trehalose and pullulan, but also ODF itself improves maintaining WIV activity, even at challenging storage conditions (60 °C/0% RH and 30 °C/56% RH).

## 4. Conclusions

ODFs with antigens (β-galactosidase or WIV) incorporated could be successfully prepared by pipetting solutions containing antigen onto plain ODFs followed by air- or vacuum-drying. The biological activity of WIV incorporated in ODFs was better preserved when formulated with sugars than without, indicating the protecting effect of sugars on WIV stability. Moreover, WIV incorporated in ODFs together with sugars showed excellent stability even when exposed to challenging storage conditions (60 °C/0% RH and 30 °C/56% RH) for 4 weeks. However, WIV air- or vacuum-dried without ODFs substantially lost its activity after 7 days of storage under the same conditions. Therefore, not only trehalose and pullulan, but also the components of the ODF (predominately HPMC) improved the stability of the WIV. Overall, this study took a step towards the development of a stable user-friendly dosage form to deliver WIV to the oral cavity.

In this research, WIV vaccine was pipetted onto ODFs. Obviously, this pipetting technique cannot be used for large-scale production. However, several industrially-applied printing techniques, e.g., 3D printing by 3D bioplotter^®^, could replace the pipetting technique enabling large-scale production.

## Figures and Tables

**Figure 1 pharmaceutics-12-00245-f001:**
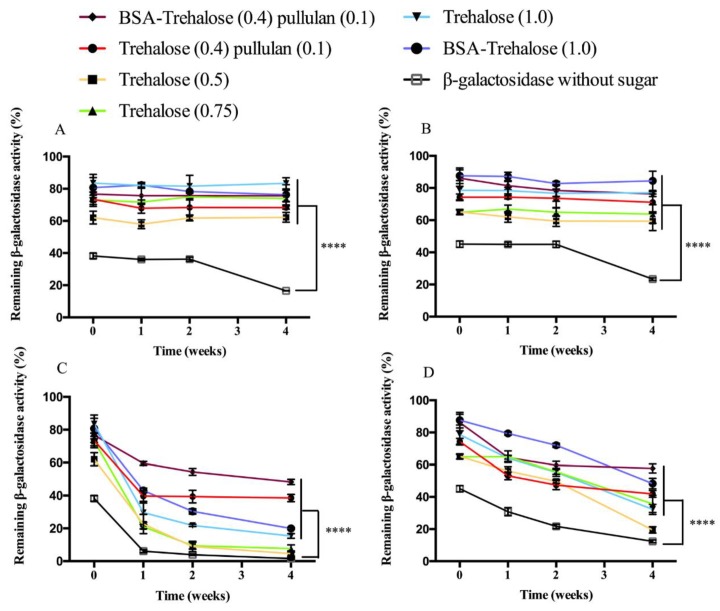
Process and storage stability of β-galactosidase incorporated in air- (**A** and **C**) and vacuum-dried (**B** and **D**) ODFs up to 4 weeks at 4 °C/0% RH (**A** and **B**) or 30 °C/0% RH (**C** and **D**). Levels of significance are denoted as **** *p* < 0.0001.

**Figure 2 pharmaceutics-12-00245-f002:**
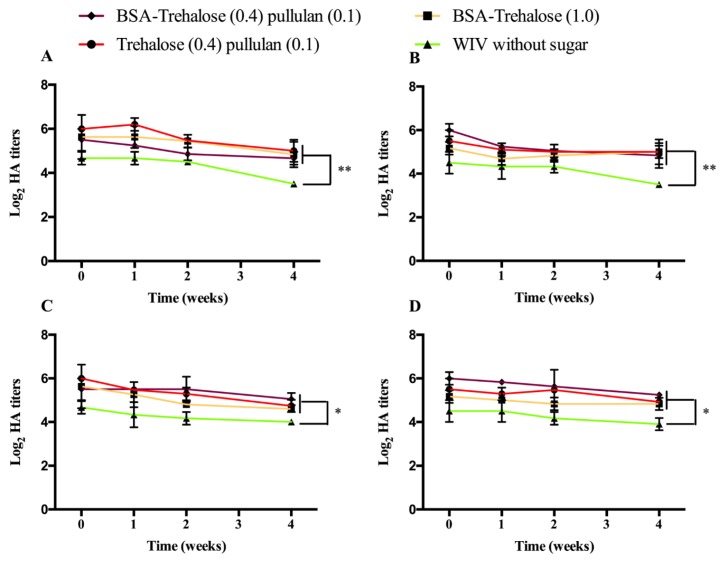
Process and storage stability of WIV incorporated in air- (**A** and **C**) and vacuum-dried (**B** and **D**) ODFs up to 4 weeks at 60 °C/0% RH (**A** and **B**) or 30 °C/56% RH (**C** and **D**). Hemagglutination titers are represented as log_2_ titers with significance indicated as * *p* < 0.05 and ** *p* < 0.01.

**Figure 3 pharmaceutics-12-00245-f003:**
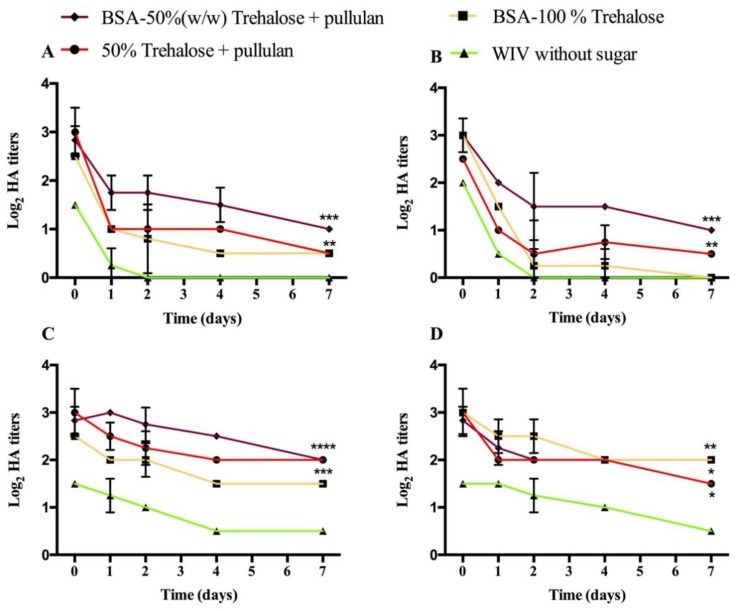
Process and storage stability of air- (**A** and **C**) and vacuum-dried (**B** and **D**) WIV dots up to 7 days at 60 °C/0% RH (**A** and **B**) or 30 °C/56% RH (**C** and **D**). Hemagglutination titers are represented as log_2_ titers with significance indicated as * *p* < 0.05. ** *p* < 0.01.*** *p* < 0.001 and **** *p* < 0.0001.

**Table 1 pharmaceutics-12-00245-t001:** Composition of β-galactosidase formulations.

Formulation Code	Trehalose (mg)	Pullulan (mg)	BSA (mg)	β-Galactosidase (mg)	Methylene Blue (mg)	Total Volume of HEPES Buffer (μL)
BSA-Trehalose (0.4) pullulan (0.1)	200	50	10	1	0.02	500
Trehalose (0.4) pullulan (0.1)	200	50	/	1	0.02	500
Trehalose (0.5)	250	/	/	1	0.02	500
Trehalose (0.75)	375	/	/	1.5	0.02	500
Trehalose (1.0)	500	/	/	2	0.02	500
BSA-Trehalose (1.0)	500	/	10	2	0.02	500
Without sugar	/	/	/	2	0.02	500

**Table 2 pharmaceutics-12-00245-t002:** Composition of WIV formulations.

Formulation Code	Trehalose (mg)	Pullulan (mg)	BSA (mg)	WIV (μg)	Methylene Blue (mg)	Total Volume of HEPES Buffer (μL, 2 mM, pH 7.4)	* Density of the Solution (mg/mL)	** WIV Content in Each Dot (6 μL) (μg)
BSA-Trehalose (0.4) pullulan (0.1)	120	30	3.2	320	0.02	300	1170	5
Trehalose (0.4) pullulan (0.1)	120	30	/	332	0.02	300	1130	5
BSA-Trehalose (1.0)	200	/	2.7	270	0.02	200	1250	5
Without sugar	/	/	/	250	0.02	300	1000	5

* The density of the solution was determined by weighing. ** WIV content was corrected by volume of the sugar solution.

**Table 3 pharmaceutics-12-00245-t003:** Mass, thickness, disintegration time, and mechanical properties of air-dried ODFs with β-galactosidase.

Sugar/β-Galactosidase Solutions	BSA-Trehalose (0.4) Pullulan (0.1)	Trehalose (0.4) Pullulan (0.1)	Trehalose (0.5)	Trehalose (0.75)	Trehalose (1.0)	BSA-Trehalose (1.0)	Without Sugar	Plain Films ^#^
Weight (mg)	33.96 ± 0.94 ****	33.82 ± 1.25 ****	34.43 ± 0.48 ****	37.67 ± 0.85 ****	38.87 ± 0.83 ****	38.83 ± 1.32 ****	26.08 ± 1.24 **	24.24 ± 1.23
Thickness (μm)	118.56 ± 5.66 ****	119.04 ± 3.49 ****	109.09 ± 3.07 ****	135.69 ± 4.35 ****	143.82 ± 5.13 ****	147.47 ± 6.03 ****	71.6 ± 2.81 ****	59.71 ± 3.25
Disintegration time (s)	22.67 ± 1.24 ****	21.40 ± 0.72 ****	23.81 ± 1.28 ****	25.48 ± 1.77 ****	26.68 ± 1.63 ****	27.31 ± 2.58 ****	16.81 ± 2.59	16.52 ± 1.66
Tensile strength (N/mm^2^)	2.37 ± 1.03 ****	1.97 ± 0.50 ****	1.47 ± 0.61 ****	1.23 ± 0.34 ****	1.17 ± 0.27 ****	1.10 ± 0.50 ****	4.80 ± 1.06	5.93 ± 1.57
Elongation at break (%)	6.24 ± 1.58 ****	6.24 ± 1.20 ****	6.14 ± 2.39 ****	5.41 ± 1.67 ****	3.5 ± 1.91 ****	4.82 ± 1.76 ****	10.72 ± 1.44 **	15.28 ± 2.84

^#^ Differences between plain films and each formulation were analyzed statistically. Levels of significance are denoted as ** *p* < 0.01, **** *p* < 0.0001.

**Table 4 pharmaceutics-12-00245-t004:** Mass, thickness, disintegration time, and mechanical properties of vacuum-dried ODFs with β-galactosidase.

Sugar/β-Galactosidase Solutions	BSA-Trehalose (0.4) Pullulan (0.1)	Trehalose (0.4) Pullulan (0.1)	Trehalose (0.5)	Trehalose (0.75)	Trehalose (1.0)	BSA-Trehalose (1.0)	Without Sugar	Plain Films ^#^
Weight (mg)	33.61 ± 0.91 ****	32.43 ± 0.81 ****	33.76 ± 0.70 ****	36.40 ± 1.19 ****	38.48 ± 2.04 ****	38.94 ± 0.43 ****	27.6 ± 0.78 ****	23.31 ± 1.32
Thickness (μm)	117.6 ± 2.37 ****	116.5 ± 3.17 ****	108 ± 5.45 ****	113.67 ± 3.81 ****	149.53 ± 7.34 ****	150.47 ± 12.78 ****	75.40 ± 2.95 ****	59.93 ± 3.65
Disintegration time (s)	21.82 ± 1.16 ****	21.99 ± 1.47 ****	22.03 ±1.06 ****	23.63 ± 1.51 ****	25.13 ± 0.82 ****	24.55 ± 1.90 ****	15.63 ± 0.79	15.87± 1.47
Tensile strength (N/mm^2^)	2.04 ± 1.25 ****	1.92 ± 1.23 ****	1.85 ± 0.80 ****	1.07 ± 0.25 ****	1.05 ± 0.45 ****	0.71 ± 0.15 ****	4.22 ± 1.58 **	5.43 ± 1.96
Elongation at break (%)	4.70 ± 2.21 ****	6.51 ± 3.53 ****	4.96 ± 1.73 ****	5.05 ± 1.03 ****	4.73 ± 1.62 ****	3.66 ± 1.83 ****	10.13 ± 1.44 **	14.59 ± 2.94

^#^ Differences between plain films and each formulation were analyzed statistically. Levels of significance are denoted as ** *p* < 0.01, **** *p* < 0.0001.

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
