# Peer review of "Development of an Orodispersible Film Containing Stabilized Influenza Vaccine"

_pharmaceutics, 2020, doi:10.3390/pharmaceutics12030245_

Round 1
Reviewer 1 Report
The authors developed a new method for stabilizing the enzyme activity of protein functions by an orodispersible film (ODF). The article reported beta-glycosidase as a model protein and then using whole inactivated virus (WIV) as an model influenza vaccine antigen. The stablization was demonstrated in the presence of sugar molecules. Hemaalgutination activity was used to determine the functionality of the WIV antigens. The manuscript can be accepted for publication with Pharmaceutics after some points have been addressed:
- The title with "vaccine delivery" seems to be over stretched as the results presentaed are on the protein stabilization. No antigenicity assay (with pAbs or mAbs) and no immunogenicity data were reported. Let alone the protectivity of the vacccine to the virus challenge. Thus itis nor appropriate using the "vaccine delivery" in the title. Instead, protein stabilization, enhanced stability, etc could be the key word in the title for this work.
- The physico-chemical characterization on the WIV should be added, such as the size, polydispersity, differential scanning calorimetry, etc....
- Rationale of choosing guinea pigs RBCs for the hemaaglutination, not from other species, should be commentted.
Minor comments:
4. Ref. 28, shouldn't the chapter or the pagination be referenced here?
Author Response
Response to Reviewer 1 Comments
We thank the editor and the reviewers for their constructive comments and questions. We have incorporated all the required additional explanations and clarifications in the revised text. Point-by-point responses to each comment are given below. We hope our manuscript is acceptable for publishing in Pharmaceutics.
Reviewer 1
The authors developed a new method for stabilizing the enzyme activity of protein functions by an orodispersible film (ODF). The article reported beta-glycosidase as a model protein and then using whole inactivated virus (WIV) as an model influenza vaccine antigen. The stablization was demonstrated in the presence of sugar molecules. Hemaalgutination activity was used to determine the functionality of the WIV antigens. The manuscript can be accepted for publication with Pharmaceutics after some points have been addressed:
- The title with "vaccine delivery" seems to be over stretched as the results presentaed are on the protein stabilization. No antigenicity assay (with pAbs or mAbs) and no immunogenicity data were reported. Let alone the protectivity of the vacccine to the virus challenge. Thus itis nor appropriate using the "vaccine delivery" in the title. Instead, protein stabilization, enhanced stability, etc could be the key word in the title for this work.
Response 1: As requested by the reviewer, we changed the title into ‘Development of an Orodispersible Film Containing Stabilized Influenza Vaccine’.
- The physico-chemical characterization on the WIV should be added, such as the size, polydispersity, differential scanning calorimetry, etc....
Response 2: We used WIV derived from the same strain as we used in previous studies (Tomar et al [1,2]). The physico-chemical properties were described in these papers and are included in the revised version of this manuscript.
- Rationale of choosing guinea pigs RBCs for the hemaaglutination, not from other species, should be commentted.
Response 3: We have a very well standardized protocol of hemagglutination and hemagglutination inhibition assay. It has been published previously that the guinea pig erythrocytes express both α-2,3-Gal and α-2,6-Gal linkages required for binding of avian and human influenza strains, respectively [3,4]. Our assay works well and gives us reliable results upon use of guinea pig RBCs for several influenza strains (human and avian). For this particular strain of the virus, the same has also been used and published before [1,2].
Minor comments: 4. Ref. 28, shouldn't the chapter or the pagination be referenced here?
Response: The text in manuscript refers to the guidance from U.S. Food and Drug Administration (FDA), this reference and text were revised.
References:
1 Tomar, J.; Patil, H.P.; Bracho, G.; Tonnis, W.F.; Frijlink, H.W.; Petrovsky, N.; Vanbever, R.; Huckriede, A.; Hinrichs, W.L.J. Advax augments B and T cell responses upon influenza vaccination via the respiratory tract and enables complete protection of mice against lethal influenza virus challenge. J. Control. Release 2018, 288, 199–211.
- Tomar, J.; Biel, C.; de Haan, C.A.M.; Rottier, P.J.M.; Petrovsky, N.; Frijlink, H.W.; Huckriede, A.; Hinrichs, W.L.J.; Peeters, B. Passive inhalation of dry powder influenza vaccine formulations completely protects chickens against H5N1 lethal viral challenge. Eur. J. Pharm. Biopharm. 2018, 133, 85–95.
- Trombetta, C.M.; Ulivieri, C.; Cox, R.J.; Remarque, E.J.; Centi, C.; Perini, D.; Piccini, G.; Rossi, S.; Marchi, S.; Montomoli, E. Impact of erythrocyte species on assays for influenza serology. J. Prev. Med. Hyg. 2018, 59, E1–E7.
- Ito, T.; Suzuki, Y.; Mitnaul, L.; Vines, A.; Kida, H.; Kawaoka, Y. Receptor specificity of influenza A viruses correlates with the agglutination of erythrocytes from different animal species. Virology 1997, 227, 493–499.

Reviewer 2 Report
This manuscript reports on the development of a stable orodispersible film for influenza vaccine delivery. The effect of sugars on the stability of the film was examined. It is interesting and describes the results carefully. However, reviewer thinks several points to be considered with detail discussion.
- In table 3 and 4, significant levels are shown as P values. But the control is not shown.
- In line 222 to 223, the authors describe the incorporation of sugars make the ODFs become more brittle. It is not clear what parameters means the description. Tensile strength or elongation at break?
- In table 3 and 4, tensile strength and elongation at break were decreased by adding sugars. Please discuss the reasons why sugars affect the mechanical properties.
- There are differences in storage stability profiles between Figure 1C and Figure 1D. Please discuss the differences between air dried OFDs and vacuum dried OFDs under the storage condition at 30°C/0% RH.
- In conclusion, the authors mention the components of the ODF itself improved the stability of the WIV. Please discuss what components contribute the improvement of the stability and what mechanism works.
Author Response
Response to Reviewer 2 Comments
We really appreciate and benefited from the constructive comments provided by both the editor and reviewer. We hope that our responses can properly address the reviewer’s concerns and the revised version is now suitable for Pharmaceutics.
Listed below are our point-by-point responses to comments from reviewer 2.
Reviewer 2
This manuscript reports on the development of a stable orodispersible film for influenza vaccine delivery. The effect of sugars on the stability of the film was examined. It is interesting and describes the results carefully. However, reviewer thinks several points to be considered with detail discussion.
- In table 3 and 4, significant levels are shown as P values. But the control is not shown.
Response 1: We thank the reviewer for noticing this. The data in Table 3 and 4 were statistically analyzed between results from each group and control group (plain films). To avoid ambiguity, an explanation was made in the manuscript.
- In line 222 to 223, the authors describe the incorporation of sugars make the ODFs become more brittle. It is not clear what parameters means the description. Tensile strength or elongation at break?
Response 2: The low tensile strength indicates the high brittleness of the films, and low elongation at break means films are not flexible. The short explanation was added in the manuscript.
- In table 3 and 4, tensile strength and elongation at break were decreased by adding sugars. Please discuss the reasons why sugars affect the mechanical properties.
Response 3: As the components in the formulation of ODFs work with forming the good plain films, which is not brittle and has high flexibility. When sugar solution applying to ODFs, those film-forming components are somehow replaced by the sugar solution pipetted onto the surface of ODFs. The high concentration of sugar, especially trehalose does not positively affect ODFs mechanical properties, on the contrary, it makes ODFs more brittle and less flexible than plain films. This phenomenon can be explained by the low molecular weight of trehalose, which makes it a poor film former. We incorporated this explanation in the manuscript.
- There are differences in storage stability profiles between Figure 1C and Figure 1D. Please discuss the differences between air dried OFDs and vacuum dried OFDs under the storage condition at 30°C/0% RH.
Response 4: Indeed, β-galactosidase incorporated air-dried ODFs and vacuum-dried ODFs showed quite different storage stability of β-galactosidase upon storage at 30°C/0% RH. Apparently, β-galactosidase was better encapsulated in vacuum-dried ODFs than air-dried ODFs. This probably can be explained by the slow drying process of vacuum drying. The vacuum-drying takes 24 h, while air-drying only needs 4 h. The text in the manuscript was revised.
- In conclusion, the authors mention the components of the ODF itself improved the stability of the WIV. Please discuss what components contribute the improvement of the stability and what mechanism works.
Response 5: The referee is right that the positive effect of ODFs components need to be further discussed. HPMC as the major component of ODFs has been showed the ability to enhance thermal stability of protein [1,2]. Therefore, we assume the predominate component HPMC contributions to the improvement of WIV stability into ODFs. In other words, WIV will not only be encapsulated by the sugars when present in the pipetting solution, but also by the ODF components (HPMC), which may affect stabilization. Furthermore, as mentioned in the manuscript, ODFs may also act as a protection layer helps to encapsulate WIV against high temperature and moisture.
References:
- Shetty, G.R.; Rao, B.L.; Asha, S.; Wang, Y.; Sangappa, Y. Preparation and characterization of silk fibroin/hydroxypropyl methyl cellulose (HPMC) blend films. Fiber. Polym. 2015, 16, 1734–1741.
- Ding, C.; Zhang, M.; Li, G. Preparation and characterization of collagen/hydroxypropyl methylcellulose (HPMC) blend film. Carbohydr. Polym. 2015, 119, 194–201.

Round 2
Reviewer 1 Report
no further comments.